# EL V.2 Model for Predicting Food Safety Risks at Taiwan Border Using the Voting-Based Ensemble Method

**DOI:** 10.3390/foods12112118

**Published:** 2023-05-24

**Authors:** Li-Ya Wu, Fang-Ming Liu, Sung-Shun Weng, Wen-Chou Lin

**Affiliations:** 1Food and Drug Administration, Ministry of Welfare, Taipei 115209, Taiwan; lywu@fda.gov.tw (L.-Y.W.); wenjou@fda.gov.tw (W.-C.L.); 2Department of Information and Finance Management, National Taipei University of Technology, Taipei 10608, Taiwan; wengss@ntut.edu.tw

**Keywords:** machine learning, ensemble learning, border management, food safety, risk prediction

## Abstract

Border management serves as a crucial control checkpoint for governments to regulate the quality and safety of imported food. In 2020, the first-generation ensemble learning prediction model (EL V.1) was introduced to Taiwan’s border food management. This model primarily assesses the risk of imported food by combining five algorithms to determine whether quality sampling should be performed on imported food at the border. In this study, a second-generation ensemble learning prediction model (EL V.2) was developed based on seven algorithms to enhance the “detection rate of unqualified cases” and improve the robustness of the model. In this study, Elastic Net was used to select the characteristic risk factors. Two algorithms were used to construct the new model: The Bagging-Gradient Boosting Machine and Bagging-Elastic Net. In addition, F_β_ was used to flexibly control the sampling rate, improving the predictive performance and robustness of the model. The chi-square test was employed to compare the efficacy of “pre-launch (2019) random sampling inspection” and “post-launch (2020–2022) model prediction sampling inspection”. For cases recommended for inspection by the ensemble learning model and subsequently inspected, the unqualified rates were 5.10%, 6.36%, and 4.39% in 2020, 2021, and 2022, respectively, which were significantly higher (*p* < 0.001) compared with the random sampling rate of 2.09% in 2019. The prediction indices established by the confusion matrix were used to further evaluate the prediction effects of EL V.1 and EL V.2, and the EL V.2 model exhibited superior predictive performance compared with EL V.1, and both models outperformed random sampling.

## 1. Introduction

Taiwan’s food supply relies heavily on imports, with a vast array of imported ingredients and products comprising a substantial portion of the population’s dietary consumption. This underscores the importance of managing imported food to protect public health and consumer rights. In Taiwan, the number of inspection applications for imported food has grown annually. Between 2011 and 2022, inspection applications increased from 419,000 batches to 723,000 batches, nearly doubling. Given the substantial volume of food imports, conducting border sampling inspections is of great significance for effectively strengthening control over high-risk products and accurately detecting substandard items.

Food risk management and control at Taiwan’s border employ a food inspection method, which can be primarily classified into two categories: review and inspection. The review is conducted in writing, comparing customs clearance data with product information. The inspection involves sampling selected batches and sending them to authorized inspection laboratories for pesticide, pigment, or heavy metal compound testing. The entire process can be completed in approximately three to seven days. According to Taiwan’s border inspection measures, inspection methods can be classified into general inspection, enhanced inspection, and batch-by-batch inspection. Generally, only 2 to 10% of the products are sampled for random inspection. However, if a single non-compliant item is detected for the same inspection applicant, origin, and product, the next import will be subject to enhanced inspection. Once an inspection application batch is designated for enhanced sampling, the random inspection method is still used but requires 20 to 50% sampling. If violations are detected again, 100% batch-by-batch inspection will be implemented [1].

The maintenance of imported food quality at the border primarily relies on the accurate detection of products that do not satisfy quality standards during sampling inspections, thereby preventing their importation. In 2020, Taiwan developed a first-generation ensemble learning prediction model (hereinafter referred to as EL V.1) for border management to identify high-risk products. Five algorithms were primarily utilized to predict the risk of products for inspection, including Decision Tree C5.0 and CART, Random Forest (RF), Logistic Regression (LR), and Naïve Bayes (NB). The detection rate of unqualified products via sampling inspection was significantly increased.

To further improve the detection rate and enhance the model’s robustness, this study aimed to construct the second-generation ensemble learning prediction model (hereinafter referred to as EL V.2). By refining the screening method for key risk factors and incorporating additional classification algorithms required for the modeling process (including Elastic Net (EN) and Gradient Boosting Machine (GBM)), the robustness of model prediction can be enhanced. With the assistance of Taiwan Food Cloud Big Data and seven machine learning algorithms for ensemble learning, the objective is to further improve the detection rate of unqualified products sampled for inspection, thereby ensuring the food safety of the population.

This study contributed to risk prediction for imported food controls at the border in several ways. First, the machine learning model for predicting the risk of imported food at the border was constructed using over ten years of data from Taiwan’s real-time food cloud while incorporating characteristic risk factors for data screening. Second, the classifier built through ensemble learning combined several classification algorithms, providing a robust prediction method and addressing the classification bias potentially occurring when using a single algorithm. Third, unqualified cases were more easily identified in border inspections when using the proposed model in view of its increased number of algorithms, suitable characteristic risk factors, and flexible adjustment of the sampling rate with F_β_ Fourth; it was found that the risk prediction model can help inspectors reduce pressure at work and overcome importer doubts regarding results from human inspections. Finally, EL V. 2 outperformed EL V.1 in predicting unqualified imported food, offering greater assurance for food safety control. (For a detailed comparison table between the abbreviations and full names of this article, please refer to Table A1 in the Appendix A.)

## 2. Literature Review

This study aims to construct a more robust risk prediction model than EL V.1, namely EV.2. To help achieve this, this section first discusses the collection of “characteristic risk factors” required for modeling. Secondly, the literature on the “selection of algorithms” is reviewed to understand the common algorithms used in the field of food. Furthermore, experts were invited to decide the algorithms to be used in the model by consensus. Finally, insights from the literature on “improvement of ensemble learning methods” were collected as references for the method design in this study.

### 2.1. Model Characteristic Risk Factors

The international application of big data in the field of food safety encompasses food safety-related monitoring, such as monitoring food additives, animal drug residues, heavy metals, allergens, and foodborne diseases, as well as providing early warnings for production, supply, and sales of products, food adulteration and fraud, and food safety incidents. The collection and integration of data can assist in the risk analysis and management of food, raw materials, and feed [2,3,4,5,6,7,8,9,10]. In 2016, Marvin et al. proposed that the Bayesian network algorithm can handle diverse big data and facilitate the understanding of driving factors related to food safety via systematic analysis, such as the impact of climate change on food quality, economy, and human behavior. Combined with the data, this algorithm can be used to predict possible food safety risk events [11]. In 2015, Bouzembrak et al. used the Rapid Alert System for Food and Feed (RASFF) of the European Union to construct a Bayesian network model to predict the types of food fraud that can occur in imported products of known food product categories and countries of origin. The findings can assist in border risk management and control and serve as an important reference for EU governments in conducting inspections and law enforcement [2,12,13].

The amount of imported food in the United States is increasing year by year. Due to limited inspection capacity, the Food and Drug Administration has divided the control of border imported food into two stages. The first stage is mainly electronic document review, with only 1% of imported food actually inspected each year. The second stage involves using the Predictive Risk-based Evaluation for Dynamic Import Compliance Targeting (PREDICT) system for risk prediction. Big data are employed to collect relevant data from products and manufacturers for evaluation, determining the risk level of imported goods. The risk factors calculated in the PREDICT system include at least four types of data, such as product risk (epidemic outbreak, recall, or adverse event), regulatory risk (specific factors of the manufacturer itself and past compliance with food safety regulations), factory inspection records of the manufacturer within three years, and historical data of the customs broker (quality analysis of data provided by the customs broker or importer within one year, such as reporting status). These data are used to screen factors related to the product itself for risk score calculation and further propose whether to conduct product sampling inspection [14].

The data sources used by the PREDICT system are mainly import alert and import notification data, domestic inspection and product tracking records, foreign factory inspections (such as equipment inspections), and identification system evaluation. Using these data, the PREDICT system can conduct data mining and analysis, enabling it to use artificial intelligence methods to predict the possible risks of imported goods and intercept them in a timely manner. This approach is undoubtedly the best for countries facing massive imports each year, which need to maintain normal export and import while still taking into account the safety and quality of goods.

Regarding the quality sampling inspection of imported food at the border, there are currently the following international experiences: The United States employs machine learning to assist in border inspection operations, while the European Union deploys methods such as Bayesian network analysis to predict factors that may cause border food risks, and then reports back to EU countries to strengthen their attention to import control. These practices demonstrate that big data applications, such as artificial intelligence and machine learning, can provide better operational quality for government border management and ensure the health and safety of the public. Therefore, this study referred to the data sources and practices of the European Union and the United States to collect risk factors and establish prediction model planning.

### 2.2. Selection of Algorithms

In recent years, ensemble learning has received great attention from researchers and has been widely applied in many fields for various purposes, such as medical diagnosis and disease prediction [15,16,17,18], Improvement of patient quality of life [19], Internet of Things (IoT) security [20,21], fault detection and error prediction for industrial processes [22,23,24], advertising and marketing [25], as well as agricultural monitoring, management, and productivity improvement [6,26]. In the food industry, it has been used for productivity improvement in food manufacturing, quality assessment and monitoring, food ingredient identification, food safety, and the quality of food delivery services (FDS). Parastar [10] developed a handheld near-infrared spectroscopy device based on ensemble learning for measuring and monitoring the authenticity of chicken meat that showed better performance in authenticity testing than common single classification methods such as partial least squares-discriminant analysis (PLS-DA), artificial neural network (ANN) and support vector machine (SVM). Using a combination of deep learning and ensemble learning techniques on milk spectral data, Neto [6] proposed a method for predicting common fraudulent milk adulterations in the dairy industry. Their method outperformed not only common statistical learning methods but also the Fourier transformed infrared spectroscopy (FTIR), which is typically used for identifying the composition of a sample in the dairy industry. Further, Adak [27] constructed a model with customer reviews of FDS using machine learning and deep learning techniques to predict customer sentiment about FDS. Based on previous studies and following consultation with experts, the following algorithms were used for constructing the new model: Decision Tree C5.0 and CART, Random Forest (RF), Logistic Regression (LR), Naïve Bayes (NB), Elastic Net (EN), and Gradient Boosting Machine (GBM). These algorithms offer interpretable approaches that are easy to understand by users, so they were adopted in ensemble learning for EL V.1 and EL V.2. Deep learning has not been included, given its low interpretability, but it may be considered in subsequent studies.

EL V.1 was constructed using five algorithms: Decision Tree C5.0 and CART, Random Forest (RF), Logistic Regression (LR), and Naïve Bayes (NB). These algorithms exhibit great interpretability and explain ability, so they were primarily used for prediction tasks with ensemble-based classification techniques. On the basis of EL V.1, this study intends to construct a model with higher predictive performance and greater computational efficiency. To reduce computation time, Elastic Net (EN) and Gradient Boosting Machine (GBM) were used to control the sampling decision within one minute for each batch of cases. The test results revealed that the computation time could be controlled within the limit using the EN and GBM. Therefore, they were integrated into the construction of EL V.2.

### 2.3. Improvement of Ensemble Learning Model

The ensemble learning model is jointly established by a group of independent machine learning classifiers, combines their respective prediction results, and implements an integration strategy to reduce the total error and improve the performance of a single classifier [28,29,30]. Each classifier may have different generalization capabilities, i.e., different inference abilities for various samples, similar to the opinions of different experts. Finally, combining the output of these individual classifiers can deliver the final classification results, significantly reducing the probability of classification errors in the results [9,30]. 

For example, Solano [31] proposed an ensemble voting model for solar radiation prediction based on machine learning algorithms. The results of the study show that the weighted average voting method based on random forest and classification boosting has superior performance and is also better than a single machine learning algorithm and other ensemble models. Chandrasekhar [32] used six algorithms (Random Forest, K-Nearest Neighbors, Logistic Regression, Naive Bayes, Gradient Boosting, and AdaBoost Classifier) for voting ensemble learning, which improved the accuracy of heart disease prediction. Alsulami [33] proposed a data mining model including three traditional algorithms (decision trees, Naive Bays, and random forests) to evaluate student e-learning data to help policy makers make informed and appropriate decisions for their institutions. These methods effectively improve model prediction performance by using three ensemble techniques, including bagging, boosting, and voting. The combination of multiple different classifiers has been proven to improve the classification accuracy of the overall classification system [34,35,36,37].

In this study, four methods proposed by scholars were utilized to enhance the diversity of classification models (or classifiers) within the ensemble learning model, including the use of different training datasets and training of different classification models with different parameter settings, algorithms, and characteristic factors [31,38]. In previous studies, five algorithms were used to construct the ensemble learning model EL V.1. To improve and stabilize the predictive performance of the model, in this study, an attempt was made to construct model EL V.2 by adding “algorithmic classification models”, adjusting the “factor screening method”, and adding “sampling rate control parameters” such that the prediction method of imported food sampling inspection at the border can play a better role. Therefore, in addition to the algorithms used in the first-generation ensemble learning model EL V.1 constructed in previous studies (including Decision Tree C5.0 and CART, Random Forest (RF), Logistic Regression (LR), and Naïve Bayes (NB)), the newly added algorithms in this study were Elastic Net (EN) and Gradient Boosting Machine (GBM). The aforementioned seven algorithms, combined with the classification model constructed by the bagging method, will use the integration method for strategic integration with the “majority decision” approach. After completing the model construction, the prediction of border inspection applications will be conducted.

## 3. Materials and Methods

To improve the robustness of EL V.2, improvements were made on the basis of EL V.1 with a more refined method for selecting characteristic risk factors and an increased number of algorithms for classification. The details of the research methodology are described in the following sections.

### 3.1. Data Sources and Analytical Tools

The modeling data for this study were sourced from the food cloud established by the Food and Drug Administration of the Ministry of Health and Welfare of Taiwan. The food cloud is centered around the Food and Drug Administration’s Five Systems, including the Registration Platform of Food Businesses System (RPFBS), the Food Traceability Management System (FTMS), the Inspection Management System (IMS), the Product Management Decision System (PMDS), and the Import Food Information System (IFIS). Additionally, it comprises cross-agency data communication, including financial and tax electronic invoices, customs electronic gate verification data, national business tax registration data, industrial and commercial registration data, indicated chemical substance flow data, domestic industrial oil flow data, imported industrial flow data, waste oil flow data, toxic chemical substance flow data, feed oil flow data, and campus food ingredient login and inspection data [39]. After imported food enters Taiwan, it must be declared and inspected through IFIS. Only after approval can the imported food enter the domestic market. The relevant business data must be registered in RPFBS, national business tax registration data, and business registration data. The flow information generated by domestic and imported products entering the market from the border should be recorded in IFIS and FTMS, as well as in electronic invoices and electronic gate goods import and export verification records. All government-conducted product sampling inspection records should be saved in PMDS, IFIS, and IMS. Information related to the company’s products can also be accessed via RPFBS and FTMS.

The main sources of this study were border inspection application data, food inspection data, food product flow information, and business registration data from Taiwan’s food cloud, as well as international open data databases related to food safety, including gross domestic product (GDP), GDP growth rate, global food security index, corruption perceptions index (CPI), human development index (HDI), legal rights index (LRI), and regional political risk index. A total of 168 factors were included in the analysis. The analytical tools used in the study were R 3.5.3, SPSS 25.0, and Microsoft Excel 2010.

### 3.2. Research Methodology

In this study, we selected food inspection application data of S-type products that had been sampled and had inspection results as the research scope. The data were divided into training, validation, and testing sets. First, different data types and analysis methods of the training set were considered to establish various models. The optimal model was selected from the prediction results obtained by importing validation set data into the model. The selected optimal model was further imported into the test set for model validation and effectiveness evaluation and confirmation, completing the construction of EL V.2.

The entire modeling process was based on previous studies on the construction of the EL V.1 method, and improvements were made to this method to aid in improving the hit rate of unqualified products detected via sampling inspection. According to the execution order, this study can be divided into four stages: “data collection”, “data integration and pre-processing”, “establishing risk prediction models”, and “evaluating prediction effectiveness”. “Establishing risk prediction models” included three procedures: “characteristic factor extraction”, “data mining and modeling”, and “establishing the optimum prediction model”. Changes were made in the calculation methods of “characteristic factor extraction” and “data mining”, as shown below: (Figure 1).

#### 3.2.1. Data Collection

The data in this study included the border inspection application database, inspection database, flow direction database, and registration database of Taiwan Food Cloud, as well as open information related to international food risk. (as shown in Table 1) A total of 168 factors were used as the main data source for constructing the risk prediction model (as shown in Table 1).

#### 3.2.2. Integration and Data Pre-Processing

In addition to data noise cleaning, the data needed to be subjected to manufacturer name and product name attribution and data string filing to further integrate the data in accordance with six aspects: manufacturer, importer, customs broker, border inspection, product, and country of manufacture. The integration process included data cleaning, error correction, and attribution.

#### 3.2.3. Establishment of Risk Prediction Model

Data processing:

This step required data segmentation by year to prepare training, validation, and test sets. The training set was divided into two forms: 2011–2017 and 2016–2017. The validation set was data from 2018, and the test set was data from 2019. To realize accurate model prediction, in this study, we first attempted to model these two data forms and then used the validation set to confirm the most suitable time interval for data modeling.

Selection of characteristic risk factors:

This step was to improve the first-generation model of EL V.1. There were two strategies for extracting characteristic factors. First, the “single-factor analysis” and “stepwise regression”, used to extract characteristic factors in EL V.1, were changed to Elastic Net. Specifically, Elastic Net is a combination of Lasso regression (i.e., L1 normalization) and Ridge regression (i.e., L2 normalization). The equations are as follows: (e.g., Equations (1)–(3))

Lasso regression:(1)$$min\sum_{i=1}^{n}V\left(f\left(x_{i}\right),y_{i}\right)+\lambda \sum_{j=1}^{p}\left|\beta \right|$$

Ridge regression:(2)$$min\sum_{i=1}^{n}V\left(f\left(x_{i}\right),y_{i}\right)+\lambda \sum_{j=1}^{p}\beta_{j}^{2}$$

Elastic Nets:(3)$$min\sum_{i=1}^{n}V\left(f\left(x_{i}\right),y_{i}\right)+\lambda_{1}\sum_{j=1}^{p}\left|\beta_{j}\right|+\lambda_{2}\sum_{j=1}^{p}\beta_{j}^{2}$$

Lasso regression can aid Elastic Net in selecting characteristic factors. When selecting variable factors, Lasso regression retains only one highly collinear variable, making it the best choice. Ridge regression filters the independent variables into separate groups such that highly collinear variables can exist in the model when they have an effect on dependent variables as opposed to retaining only one of them, like in Lasso regression. Ogutu et al. indicated that due to its own characteristics, Elastic Net would try its best to discard variables within the model that have no influence on the independent variables, which can improve the explanatory power and predictive capability of the model. Relatively speaking, if all highly collinear independent variable factors are retained, the prediction performance of the model may not be increased, and the model will become more complex and unstable [40]. In this study, there were many factors. Hence, there were doubts about high collinearity. To avoid the problem of collinearity among factors that may be ignored when using “single-factor analysis and stepwise regression” to select factors in the past, Elastic Net was selected to reduce the possible bias of the prediction model and improve the accuracy of prediction.

The second strategy involved modeling based on inspection data from 2011 to 2017. Monthly data from January to October 2018 were added over time. The model was updated once a month, and the number of characteristic factors used was calculated. With seven algorithms, each factor can be used up to 70 times. The factor that was used more than once was kept and included in the model required for EL V.2 construction. In this study, a total of 68 characteristic risk factors were obtained (as shown in Table 2), which were important characteristic factors that participated in EL V.2 modeling.

Data exploration and modeling

In this study, we conducted modeling based on the training set. In addition to the algorithms used in EL V.1 (including Bagging-C5.0, Bagging-CART, Bagging-LR, Bagging-RF, and Bagging-BN), Bagging-EN and Bagging-GBM were also added for “data mining and modeling”. Bagging can train multiple prediction classifiers for the same algorithm with a non-weighted method, which is then aggregated into the model constructed by the computational classifier. In this study, we used seven models established by Bagging-C5.0, Bagging-CART, Bagging-LR, Bagging-RF, Bagging-BN, Bagging-EN, and Bagging-GBM, and then ensembled them via the voting rule of “majority decision” as the final ensemble prediction model (Figure 2).

Establish the optimum prediction model○Training set resamplingAccording to historical border inspection application data, the number of unqualified batches accounts for a small proportion of the total number of inspection applications, and modeling based on this data can easily lead to prediction bias. Therefore, in this study, we adopted two resampling methods (the synthesized minority oversampling technique (SMOTE) and proportional amplification) to deal with the data imbalance problem and tried to use the ratios of qualified to unqualified batches of 7:3, 6:4, 5:5, 4:6, and 3:7 for evaluation to find the best proportional parameters and unbalanced data processing method.○Repeated modelingIn this study, after the training set was resampled to balance the number of qualified and unqualified cases, the data combination of “time interval (AD)/whether to include the vendor blacklist/data imbalance processing method” was used to reduce the misjudgment due to a single sampling error. There were two types of time intervals (AD): 2016–2017 and 2016–2017. Blacklisted vendors refer to those whose unqualified rate was greater than the average of the overall unqualified rate. The most commonly used methods for handling data imbalance were proportional amplification and SMOTE. Based on this combination, a total of six types A to F were formed, namely, A: 2016–2017/Yes/Proportional Amplification, B: 2016–2017/Yes/SMOTE, C: 2011–2017/Yes/Proportional Amplification, D: 2011–2017/Yes/SMOTE, E: 2011–2017/No/Proportional Amplification, and F: 2011–2017/No/SMOTE. Repeated modeling was conducted ten times, and the average was used to establish the model.○Selection of the optimal modelThe validation data set was imported into the model to obtain seven classifiers established by seven algorithms. Then seven classifiers were integrated for integrated learning to extract the optimum prediction model from the predicted results.

#### 3.2.4. Evaluation of the Prediction Effectiveness

In this step, the test set was imported into the model, and the confusion matrix (Table 3) output prediction indicators (accuracy rate (ACR), F1, positive predictive value (PPV), Recall, and area under curve (AUC) of receiver operating characteristic (ROC)) were used to evaluate the prediction effect. The purpose was to confirm whether the model can improve the predictive effect of the unqualified rate for border inspection applications.

ACR represents the model’s ability to discriminate among overall samples. However, due to the presence of unbalanced samples in this study and the small number of unqualified samples, ACR may tend to present qualified prediction results due to its strong discriminative power towards qualified predictions. Therefore, in this study, more emphasis was placed on PPV, Recall, and F1 (Equation (4)). Recall represents the proportion of the number of unqualified products correctly identified by the model to the total number of unqualified products (Equation (5)). PPV refers to the proportion of the number of products that are actually unqualified to the number of products identified by the model as unqualified, making it also known as the unqualified rate (Equation (6)). F1 is the harmonic mean of recall and positive predictive value. Assuming that the PPV and F1 thresholds are set to 0.5, i.e., the weights of the two are equal, the performance of F1 is estimated. The larger the numerical value, the more favorable it is for the number of unqualified products TP to increase (Equation (7)).
ACR = (TP + TN)/(TP + TN + FP + FN) (4)
Recall = TP/(FN + TP) (5)
PPV = TP/(TP + FP) (6)
F1 = 2 (PPV × Recall)/(PPV + Recall) = 2TP/(2TP + FP + FN) (7)

The ROC can be plotted as a curve. The larger the area below the curve, the higher the classification accuracy. Performance can be compared between multiple ROC curves. The area under the curve (AUC) refers to the ratio of the area under the ROC curve divided by the total area. AUC can serve as the decision threshold when comparing the changes between the True Positive Rate (TPR) (Equation (8)) and False Positive Rate (FPR) (Equation (9)). The ROC curve is a graphical representation of a binary classification model’s performance that clarifies the trade-off between the True Positive Rate (TPR) and the False Positive Rate (FPR) for various threshold values. When TPR is equivalent to FPR, AUC = 0.5, which indicates that the results of the prediction model sampling inspection are equivalent to those of random sampling inspection, and the prediction model has no classification capability. AUC = 1 indicates that the classifier is perfect; 0.5 < AUC < 1 indicates that the model is superior to random sampling; AUC < 0.5 indicates that the model is inferior to random sampling (Figure 3).
(8)$$\mathrm{True}\;\mathrm{Positive}\;\mathrm{Rate},\;\mathrm{TPR}=\frac{TP}{TP+FN}$$
(9)$$\mathrm{False}\;\mathrm{Positive}\;\mathrm{Rate},\;\mathrm{FPR}=\frac{FP}{TN+FP}$$

The evaluation index for the effectiveness of model prediction in this study was the confusion matrix. Firstly, the classification prediction results were calculated, and the selection of models with a decision threshold greater than 0.5 for AUC (equivalent to random sampling) was prioritized. Then, a comprehensive evaluation was conducted. This study primarily focused on the unqualified rate to truly reflect the prediction hit rate. Therefore, the main evaluation index was the positive predictive value (PPV), also known as precision, which represented the ratio of the number of samples judged as unqualified by the model to the actual number of unqualified samples. Additionally, there was Recall, which was the ratio of the number of unqualified products correctly identified by the model to the total number of unqualified products. However, the larger the Recall, the higher the sampling rate. Hence, increasing PPV within the tolerable range of the sampling rate was the most important step. This also indicated the importance of realizing a balance between the harmonic mean F1, Recall, and PPV.

#### 3.2.5. Evaluation of the Prediction Effectiveness

In this study, the data from the 2019 test set was used to make predictions through the model and simulated the actual prediction after the model launch for effectiveness evaluation. The evaluation of prediction effectiveness and selection of the optimum prediction model was based on the confusion matrix. The evaluation indicator PPV referred to the proportion of the number of products that were actually unqualified to the number of products identified by the model as unqualified. Recall referred to the accuracy of classification for all unqualified samples. EL V.1 was officially launched to conduct online risk forecasting at the border on 8 April 2020. It was switched to EL V.2 on 3 August 2020 for continuous online real-time forecasting. Therefore, in this study, we compared the unqualified rates in 2020, 2021, and 2022 after the launch with that in 2019 before the launch. The chi-square test was used to evaluate whether there was a significant increase in the unqualified rate with the aid of risk prediction and sampling of EL V.2 constructed in this study, which was used as the final evaluation result of the prediction effectiveness.

## 4. Results

### 4.1. Resampling Method and Optimal Ratio

To overcome the problem of the number of unqualified batches being too small, in this study, we tried using proportional amplification and the synthesized minority oversampling technique (SMOTE) for resampling to select the best method to deal with unbalanced data and avoid deviation in model prediction. To explore the proportional parameter of qualified to unqualified batches, tests were conducted using proportional amplification at 7:3 and SMOTE at 7:3, 6:4, 5:5, 4:6, and 3:7. After pairing with Bagging, 10 iterations were conducted to obtain the average result for each of the seven algorithms. Then, the “majority decision” in the ensemble learning method was used to obtain the results. The predictive effect was observed via PPV and F1. Previous studies found that 10 and 100 iterations of modeling exhibited comparable results, but the time required for 100 iterations significantly exceeded that for 10 iterations and was 3–8 times longer. Therefore, 10 iterations were selected for modeling, considering the time limitations.

In this study, we selected the inspection data of S-type food as the training set. After ensemble learning, the research results showed (Table 4) that when the extracted PPV and F1 were the highest, the optimal proportion of imbalanced sample processing was SMOTE 7:3. F1 was 11.03%, PPV was 6.03%, and Recall was 64.91%. Therefore, this study adopted a 7:3 ratio for qualified to unqualified samples. Based on historical experience, a ratio of 7:3 was used for proportional amplification in this study. It was not yet confirmed that SMOTE and proportional amplification were the most suitable methods for processing imbalanced data in this study. Therefore, both will continue to be included in the evaluation project in the future.

### 4.2. Generation of the Optimum Prediction Model

In this study, the “time interval” and “whether blacklisted manufacturers were included” were used as fixed risk factors in the training set, and the unbalanced data processing method of “SMOTE or proportional amplification” was adopted. Therefore, six data combinations were generated in the study, named A–F. Subsequently, seven algorithms were adopted for modeling, including Bagging-CART, Bagging-C5.0, Bagging-LR, Bagging-NB, Bagging-RF, Bagging-EN, and Bagging-GRM. After that, together with ensemble learning (EL), a total of 42 models and performance indicator evaluation results were generated, as listed in Table 5.

To construct the optimal prediction model in this study, the first step was to examine the effectiveness evaluation index AUC of the model, which should be greater than 50%, to ensure that the probability of unqualified batches being selected was greater than that of random sampling. Secondly, the top three combinations with the highest F1 values were prioritized. Furthermore, 25.0% for D7 random forest and both 23.0% for C8 and D8 ensemble learning indicated better performance. Another important evaluation indicator of PPV was further observed. Among the three aforementioned methods, 22.9% for the C8 ensemble method was the best. Meanwhile, Recall was 29.0%, 23.2%, and 28.3%, respectively, all of which reached the acceptable level. To comply with the requirement in practice that the general sampling rate should be controlled between 2% and 10%, it was important to note that the performance of Recalls was closely related to the sampling rate. When Recall was higher, the sampling rate was also relatively higher. Additionally, in this study, we also focused on the comparison of the number of unqualified pieces in the sampling to avoid situations where the unqualified rate was high while the sampling rate and the number of unqualified pieces were low. In summary, in this study, we selected the “C8 ensemble method” as the optimum prediction model.

In this study, we obtained similar results when examining the robustness of the model’s future prediction and the top three F1 scores of D7, C8, and D8. Therefore, a total of 16 combinations of Group C and Group D were retained for subsequent real-world prediction simulation to determine the appropriateness of the selected optimal prediction model.

### 4.3. Model Prediction Effectiveness

In this study, we imported the test set data into the best model C8 identified in the previous stage and simultaneously into combinations with similar evaluation results (including C1–7 and D1–8) to observe the predictive performance of the model. The research results showed (Table 6) that the top three models (C8, D7, and D8), which were originally the best choices, output F1 scores of 21.6%, 14.3%, and 15.8% and PPV values of 16.4%, 10.4%, and 12.3%, respectively, after the test set was imported for effectiveness evaluation. This result confirmed that C8 remained the optimum prediction model.

Table 6 demonstrates that the ensemble method for Group C (F1 21.6%, PPV 16.4%) exhibits significant or equivalent predictive results when compared to other single algorithms (C1–7: F1 3.4–22.2%, PPV 4.6–18.8%). The Group D ensemble method (F1 15.1%, PPV 12.3%) also exhibited similar prediction results as Group C when compared to seven algorithms (C1–7: F1 3.4–22.2%, PPV 4.6–18.8%; D1–7: F1 8.6–14.6%, PPV 4.7–13.0%). Therefore, compared to any other algorithm, the ensemble method in this study can have an equivalent or better effect, and it was also more robust.

In 2019, the total number of inspection batches for S-type food was 29,573, and the actual number of randomly selected batches with inspection results was 4154 (excluding annual inspection batches). These 318 batches with sampling results were used as test sets for prediction. The number of batches sampled according to the prediction model recommendation was 318. The recommended sampling rate by the model was 7.66%, the hit rate was 16.35%, and the number of hit batches of the model was 52. The original overall sampling rate was 10.68%, the unqualified rate was 2.09%, and the number of unqualified batches was 618. The hit rate of sampling inspection with model recommendation was 7.82 times that of the original random sampling (Table 7).

In summary, the results of this study showed that the C8 ensemble method was the optimal model choice for this study. After effectiveness evaluation, it was determined that the hit rate of sampling inspection after the model recommendation was greater than that of random sampling.

## 5. Discussion

To enhance the prediction performance of EL V.2, in this study, we employed several methods that differed from EL V.1. These methods included adjusting the selection approach for characteristic risk factors, incorporating additional algorithms into the model, and utilizing F adjustment to maintain the sampling rate within 2–8% after EL V.2 was launched. Simultaneously, 2% was reserved for random sampling to avoid model overfitting, thereby strengthening the robustness and prediction hit rate of ensemble model prediction results (Table 8).

### 5.1. F_β_ Was Employed to Regulate the Sampling Inspection Rate

In this study, it was discovered that during the operation of EL V.1, the risk score distribution for each model varied (Figure 4). Hence, using the same threshold F_β_ to regulate the sampling rate was not advisable. Therefore, the optimal threshold F_β_ was set for each model separately through β. The F-value employed in the current evaluation model was the harmonic mean of PPV (unqualified rate in sampling inspection) and Recall (identification rate of unqualified products in sampling inspection). F_β_ adjusted the weights of PPV and Recall based on different β values. The larger the β, the greater the weight of Recall (Equation (10)). Then, based on the threshold setting, the unqualified rate and sampling rate were evaluated.
(10)$$F_{\beta }=\left(1+\beta^{2}\right)\times \frac{PPV\times Recall}{\left(\beta^{2}\times PPV\right)+Recall}$$

In this study, we used F_β_ to identify the prediction results of the optimal threshold for each model to maximize the F value with different β values. We reviewed the model thresholds F_β_ established via various algorithms to evaluate the sampling unqualified rate and sampling rate of S-type products from 1 May 2020 to 31 May 2020. The final output is listed in the threshold regulation analysis table with different β values, as presented in Table 9.

To control the sampling rate at 7%, using Beta 2.6 as an example, the unqualified rate of sampling was 16.67%, and the sampling rate was 7.23%. When all classification models utilized the same threshold, the unqualified rate of sampling was 15.45%, and the sampling rate was 7.56% (Table 10). This study found that regulating the sampling rate with Beta can increase the unqualified rate of sampling. If the sampling rate was low, the Beta value could be adjusted higher to improve the sampling rate; if the sampling rate was too high, the Beta value could be lowered to reduce the sampling rate. Therefore, the EL V.2 constructed in this study was designed to regulate the Beta value according to the required sampling rate. Through the automated generation of optimal thresholds by the model, the accuracy of each model can be enhanced, and the effectiveness of sampling management can be strengthened.

### 5.2. Comparison between Single Algorithm and Ensemble Algorithm

Among the 42 prediction models established in the stage of optimal model selection, for each of the six data combinations of A-F, both F1 and PPV of the ensemble learning method ranked in the top three among the eight models when compared to the single algorithm. Moreover, their AUCs were all greater than that of 50% random sampling (Table 4). When further using the test set to simulate actual predictions, the ensemble method in the C and D data combinations (Table 5) remained in the top three (C8 ensemble method F1 21.6%, PPV 16.4%, AUC 69.9% > 50%; D8 ensemble method F1 15.1%, PPV 12.3%, AUC 69.0% > 50%). The results of this study showed that the ensemble method was the most suitable approach for constructing border food prediction models, and its robustness could ensure that high-risk products could be efficiently predicted and detected as unqualified through sampling and inspection. Thus, the occurrence of food safety incidents could be prevented.

### 5.3. Comparison of Prediction Effectiveness between EL V.2 and EL V.1 Models

In this section, we explored whether the second-generation ensemble learning prediction model (EL V.2) constructed by our research institute (composed of seven algorithms: Bagging-CART, Bagging-C5.0, Bagging-Logistic, Bagging-NB, Bagging-RF, Bagging-EN, and Bagging-GRM) exhibited better predictive performance than the first-generation model (EL V.1) constructed by the previous study using five algorithms: Bagging-CART, Bagging-C5.0, Bagging-Logistic, Bagging-NB, and Bagging-RF. In this study, we selected the time interval in 2020 with ensemble learning for effectiveness evaluation. EL V.1 analysis interval: 8 April 2020 to 2 August 2020; EL V.2 analysis interval: 3 August 2020 to 30 November 2020. After using the prediction index established by the confusion matrix, the results showed that:The AUC of EL V.1 ranged from 53.43% to 69.03%, while the AUC of EL V.2 ranged from 49.40% to 63.39%. After a majority decision, the Bagging-CART model of EL V.2 with AUC less than 50% was considered unsuitable. By adopting a majority decision strategy through ensemble learning, the influence of the Bagging-CART model was diluted by the other six models. Thus, EL V.2 exhibited better robustness than EL V.1. The advantage of ensemble learning was that when a small number of algorithms were not suitable (worse than random sampling), there was a mechanism for eliminating or weakening influence. The performance of AUC showed that EL V.1 and EL V.2 had a greater prediction probability than randomly selecting unqualified cases (Table 11).The predictive evaluation index F1 (8.14%) and PPV (4.38%) of EL V.2 had better results compared to F1 (4.49%) and PPV (2.47%) of EL V.1, indicating that EL V.2 had better predictive effects than EL V.1 (Table 12).

The above results indicated that EL V.2 had better predictive performance than EL V.1, but it should still be noted that the Recall of EL V.2 was about twice that of EL V.1. This suggested that there might be a relative increase in the sampling rate. Therefore, determining how to control the sampling rate within the general sampling rate range (2–10%) while improving the unqualified hit rate was a key consideration after the model’s launch.

### 5.4. Evaluation of the Effectiveness of the Prediction Model after Its Launch

In this study, we used the ensemble learning method to construct the EL V.1 model, which was launched on 8 April 2020. The S-type food was imported for sampling inspection prediction. On 3 August 2020, EL V.1 was replaced by EL V.2. To understand the effectiveness of the model after its launch, the performance from 2020 to 2022 was compared with that of the random sampling method in 2019. The results showed that from 2020 to 2022, after conducting general sampling inspection predictions using the ensemble learning model, the unqualified rates obtained were 5.10%, 6.36%, and 4.39%, respectively, which were higher than the unqualified rate of 2.09% in 2019. The overall annual sampling rates were 6.07% in 2020, 9.14% in 2021, and 10.9% in 2022, which were all controlled within the range of 2–10% (without rounding below the decimal point) (Table 13 and Table 14). In this study, we further utilized statistical analysis for the chi-square test. The results showed that the ensemble learning method for border food sampling inspection had statistical significance (*p* value = 0.000 ***) in improving the unqualified rate (Table 14). Therefore, the ensemble learning model EL V.2, constructed by the seven algorithms used in this study and launched on 3 August 2020, can effectively increase the unqualified rate while maintaining the general sampling rate within a reasonable range of 2–10%.

The findings of this study are as follows:EL V.2 is better than random sampling. After the ensemble learning model EL V.2, developed in this study, was launched online, the predicted results from 2020 to 2022 were reviewed. Based on the overall general sampling cases throughout the year, it was determined that the unqualified rate was 3.74% in 2020, 4.16% in 2021, and 3.01% in 2022, all of which were significantly higher than 2.09% in 2019. Further observation showed that the unqualified rates of cases recommended for sampling inspection through ensemble learning in 2020, 2021, and 2022 were 5.10%, 6.36%, and 4.39%, respectively, which were significantly higher than the 2.09% under random sampling inspection in 2019.The ensemble learning model should be periodically re-modeled. Based on Table 12, it can be observed that the unqualified rate showed a growing trend from 2019 to 2021 but a slight decrease in 2022 (Figure 5). The results of the further chi-square test showed that the unqualified rate in 2022 was still significantly higher than that in 2019 (*p* value = 0.000 *** < 0.001) (Table 14). However, for ensemble learning prediction models constructed using various machine learning algorithms, the factors and data required for modeling often change with factors such as the external environment and policies. Re-modeling was necessary to make the best adjustments to “data drift” or “concept drift” in the real world to prevent model failure. Drift refers to the degradation of predictive performance over time due to hidden external environmental factors. Due to the fact that data changed over time, the model’s capability to make accurate predictions may decrease. Therefore, it was necessary to monitor data drift and conduct timely reviews of modeling factors. When collecting new data, the data predicted by the model should be avoided to prevent the new model from overfitting when making predictions. The goal of this study is to enable the new model to adjust to changes in the external environment, which will be a sustained effort in the future.The trade-off between unqualified batch hit rate and computational efficiency needs to be established. While the rejection rate was improved using the model constructed with seven algorithms (i.e., EL V.2), there were approximately 0.1% of batches where the model took more than one minute to compute. The model was designed to facilitate inspectors at the border to make fast decisions on sampling. Considering computational efficiency and real-time prediction, random sampling would be automatically selected for batches with over 1 min computation time to avoid delay in border inspections due to model failure.

### 5.5. Research Limitations

When determining the research scope, it was necessary to ensure that each product classification for border inspection applications had unqualified cases and that the number of unqualified cases was not too small. Therefore, for those with an unqualified rate of less than 1% in past sampling and fewer than 10 unqualified cases, the original random sampling mechanism was maintained. The product classification was not included in the scope of this study when it was impossible to find a classification with high product homogeneity and similar inspection items that could be merged. Owing to legal requirements associated with government data, the types, content, and hyperparameters of risk factors cannot be presented in this paper to protect information security and confidentiality.

## 6. Conclusions

In this study, we constructed a second-generation integrated learning prediction model, EL V.2. The research results showed that EL V.2 exhibited better prediction performance than random sampling and the first-generation integrated learning prediction model, EL V.1. Additionally, the model was composed of seven algorithms. Hence, when the model was inadequate (AUC < 50%), the overall prediction results remained robust when integrated learning was conducted through the majority decision voting method.

The outbreak of the COVID-19 pandemic in late 2020 had a worldwide impact on border control measures as well as economic and trade exchanges. Compared with unqualified rates in 2019, 2020 and 2021 saw increases in unqualified cases in Taiwan, which is likely to be attributed to the great changes in the origin and quantity of imported goods caused by the pandemic. Another reason for the changes in unqualified rates could be the modification of some related regulations and inspection standards. The effects of these aspects on the evaluation of the performance of EL V.1 and EL V.2 still require further observation and analysis in the future. Since 2020, Taiwan’s border management has gradually introduced an intelligent management operation model. Border management powered by artificial intelligence enables Taiwan to strengthen its risk prediction capabilities and quickly adapt to trends in the context of rapid changes in the international environment, thereby ensuring people’s health and safety.

## Figures and Tables

**Figure 1 foods-12-02118-f001:**
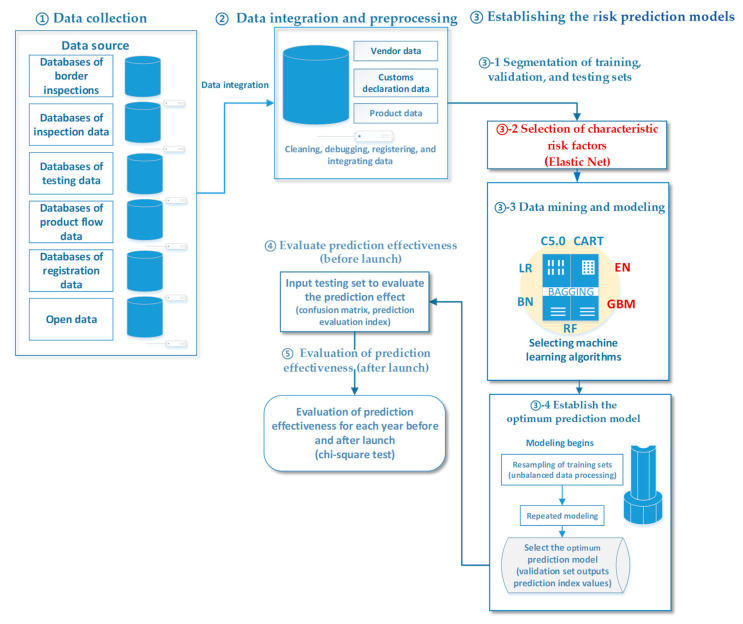
Modeling process of the second-generation ensemble learning prediction model EL V.2 (Note: The red letter indicates the difference between EL V.2 and EL V.1 modeling processes).

**Figure 2 foods-12-02118-f002:**
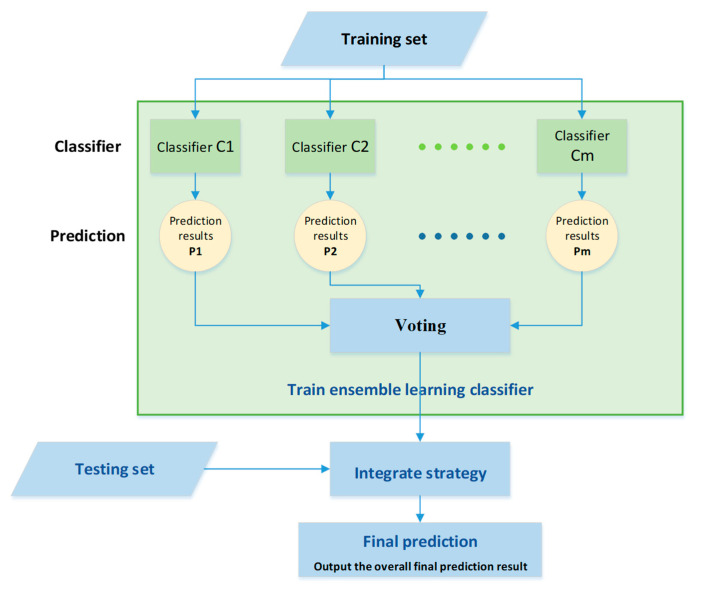
Ensemble learning model architecture.

**Figure 3 foods-12-02118-f003:**
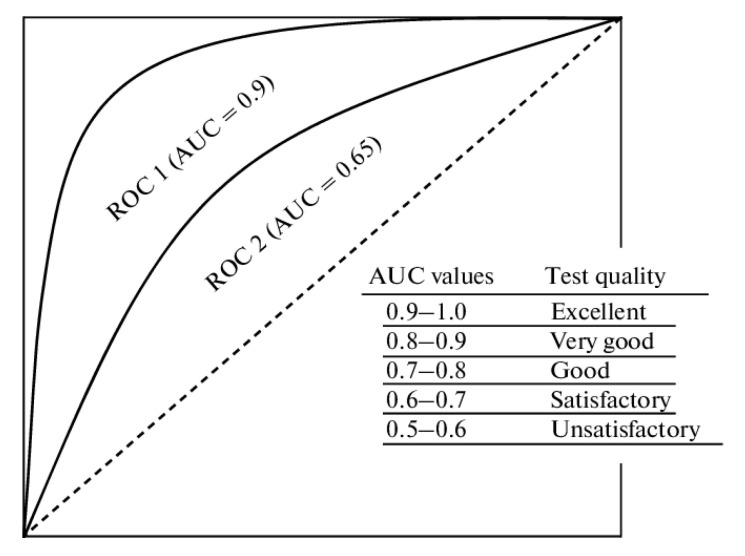
ROC curve and AUC concept map.

**Figure 4 foods-12-02118-f004:**
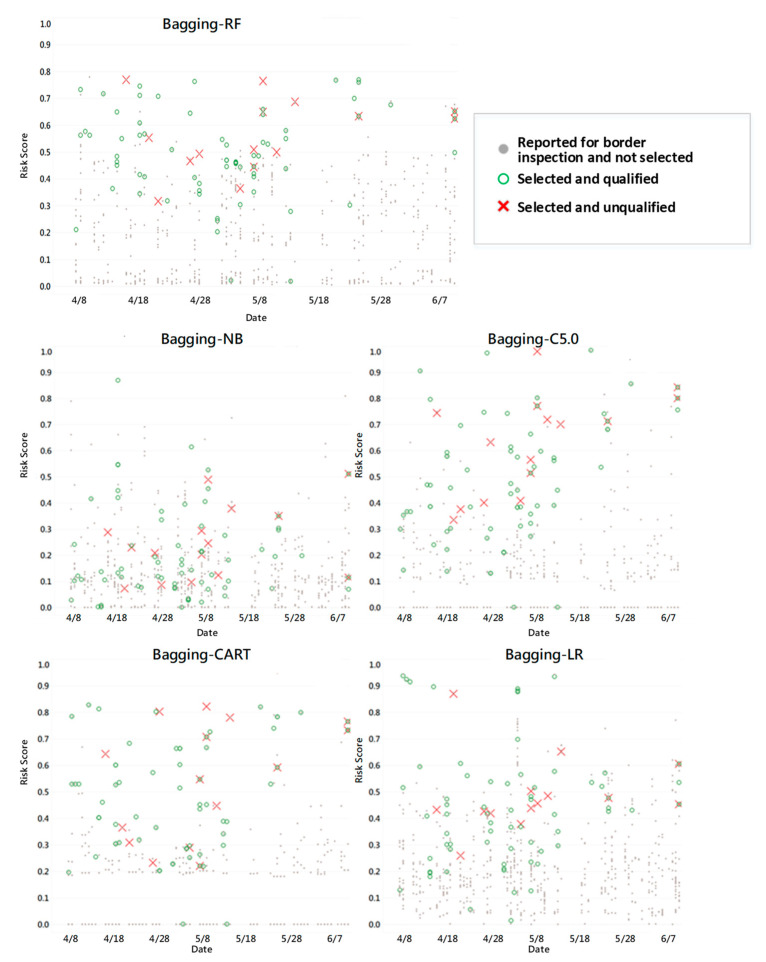
Predicted risk score distribution map of inspection application cases with five algorithms in EL V.1 as examples (Data time interval from 8 April 2020 to 7 June 2020).

**Figure 5 foods-12-02118-f005:**
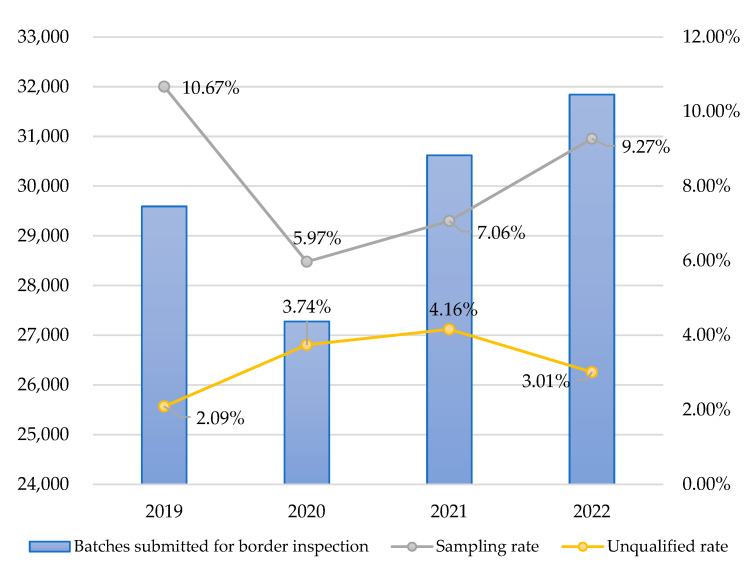
Annual trend chart before and after the introduction of ensemble learning.

**Table 1 foods-12-02118-t001:** Type and sources of characteristic factors.

Type	Factors	Data Sources
**Product**	Value, net weight, inspection methods, blacklisted products, packaging methods, validity period, products for which international recall alerts have been issued, manufacturing date, expiry date, etc.	Taiwan Food Cloud: Data on border inspections Product inspection and testing data Product alerts Information on international public opinion and product recall alerts: United States Food and Drug Administration (US FDA) https://www.fda.gov Food Safety and Inspection Service (FSIS) of the U.S. Department of Agriculture (USDA) https://www.fsis.usda.gov Rapid Alert System for Food and Feed (RASFF) of the European Union https://ec.europa.eu/food/safety/rasff_en Canadian Food Inspection Agency (CFIA) http://inspection.gc.ca Food Standards Agency (FSA) of the United Kingdom https://www.food.gov.uk Food Safety Authority of Ireland (FSAI) https://www.fsai.ie Food Standards Australia New Zealand (FSANZ) http://www.foodstandards.gov.au Consumer Affairs Agency (CAA) of Japan https://www.recall.caa.go.jp Singapore Food Agency (SFA) https://www.sfa.gov.sg China Food and Drug Administration (CFDA) http://gkml.samr.gov.cn Foodmate Network of China http://news.foodmate.net Centre for Food Safety (CFS) of Hong Kong http://www.cfs.gov.hk
Border inspection	Transportation time, month of inspection, quarter of inspection, year of inspection, method of transportation, agent importation, re-exportation, customs district, etc.	Taiwan Food Cloud: Management data of border inspections
Customs broker	Number of declarations filed, number of border inspection cancellations, number of days from the previous importation, rate of change of number of days taken for importation, number of cases of non-conforming labels and external appearances, number of batches forfeited or returned, number of inspections, number of failed inspections, number of failed document reviews, number of product classes, etc.	Taiwan Food Cloud: Food company registration data Data on border inspections Business registration data
Importer	Capital, years of establishment, number of branches, number of downstream vendors, number of company registration changes, number of late deliveries, sole focus on importation (yes/no), number of lines of businesses, new company (yes/no), district of registration, branch company (yes/no), blacklisted importer (yes/no), county/city, number of preliminary inspections, GHP inspections, HACCP inspections, label inspections, product inspections, number of lines of food businesses, factory registration (yes/no), delayed declaration of goods receipt/delivery (yes/no), interval between importations, variations in the interval between importations, variations in the number of days taken for importation, variations in total net weight, number of declarations filed, number of cases of non-conforming Chinese labels and external appearances, value, net weight, number of non-releases, number of batches detained, forfeited or returned, number of failed inspections, number of inspections, number of failed document reviews, number of border inspection cancellations, number of manufacturers, number of product classes for which declarations have been filed, total number of classes, etc.	Taiwan Food Cloud: Food company registration data Data on border inspections Product inspection and testing data Product flow data Business registration data
Manufacturer	Trademarks, interval between importations, rate of change of interval between importations, internationally alerted manufacturer (yes/no), internationally alerted brand (yes/no), number of cases of non-conforming Chinese labels and external appearances, number of batches detained, forfeited or returned, number of failed inspections, number of inspections, number of failed document reviews, number of declarations filed, number of border inspection cancellations, number of importers, number of product classes, etc.	Taiwan Food Cloud: Food company registration data Data on border inspections Product inspection and testing data Product alerts Information on international public opinion and product recall alerts: USFDA https://www.fda.gov FSIS https://www.fsis.usda.gov CFIA http://inspection.gc.ca FSA https://www.food.gov.uk RASFF https://ec.europa.eu/food/safety/rasff_en FSAI https://www.fsai.ie FSANZ http://www.foodstandards.gov.au CAA https://www.recall.caa.go.jp SFA https://www.sfa.gov.sg CFDA http://gkml.samr.gov.cn Foodmate Network of China http://news.foodmate.net CFS http://www.cfs.gov.hk
Country of manufacture	Country of manufacture of products subjected to inspection	Data on border inspections
	GDP, economic growth rate, GFSI, CPI, HDI, LRI, regional PRI	Information on international public opinion and product recall alerts: https://data.oecd.org/gdp/gross-domestic-product-gdp.htm https://www.imf.org/en/Publications https://foodsecurityindex.eiu.com/ https://www.transparency.org/en/cpi/2020/index/nzl http://hdr.undp.org/en/2020-report https://data.worldbank.org/indicator/IC.LGL.CRED.XQ https://www.prsgroup.com/regional-political-risk-index/

**Table 2 foods-12-02118-t002:** Characteristic factor usage frequency counting table.

Characteristic Factor	Times	Characteristic Factor	Times	Characteristic Factor	Times	Characteristic Factor	Times
Country of production	70	Declaration acceptance unit	17	Whether there is a trademark	5	Frequency of business registration changes	1
Inspection method	64	Advance release	17	Product registration location	5	Is it an import broker?	1
Product classification code	64	Cumulative sampling number of imports	17	Regional political risk index	5	Average price per kilogram	1
Blacklist vendor	47	Human development index	17	Tax registration data available?	5	Acceptance month	1
Cumulative number of unqualified imports in sampling inspection	45	Packaging method	15	Non-punctual declaration rate of delivery	5	Acceptance season	1
Dutiable price in Taiwan dollars	44	Capital	14	Input/output mode	4	Acceptance year	1
Total net weight	35	Import cumulative number of new classifications	13	Percentage of remaining validity period for acceptance	4	Overdue delivery	1
Global food security indicator	29	Years of importer establishment	10	Whether there is factory registration?	4	Any business registration change?	1
Type of Obligatory inspection applicant	26	Number of companies in the same group	10	New company established within the past three months	3	Product classification code	1
Legal rights index	26	Is it a pure input industry?	10	Number of GHP inspection	3	Number of GHP inspection failures	1
Blacklist product	25	Customs classification	10	Accumulated number of unqualified imports	3	Number of HACCP inspection	1
Storage and transportation conditions	23	County and city level	10	Transportation time	2	Number of HACCP inspection	1
Packaging materials	21	Total number of imported product lines	8	GDP growth rate	2	Manufacturing date later than effective date	1
Cumulative number of reports	20	Number of downstream manufacturers	7	Number of branch companies	2	Valid for more than 5 years	1
Total classification number of imports	20	Is it a branch company?	7	Number of overdue deliveries	2	Acceptance date later than manufacturing date	1
Corruption perceptions index	18	Number of non-review inspections	7	Any intermediary trade?	2	Acceptance date later than the effective date	1
GDP	17	Rate of non-timely declaration of goods received	7	Cumulative number of imports not released	2	Number of business projects in the food industry	1

**Table 3 foods-12-02118-t003:** Types and definitions of confusion matrices.

Type	Definition
True Positive, TP	Each batch of inspection applications was predicted as unqualified by the model, and it was actually unqualified.
False Positive, FP	Each batch of inspection applications was predicted as unqualified by the model, but it was actually qualified.
True Negative	Each batch of inspection applications was predicted as qualified by the model, and it was actually qualified.
False Negative	Each batch of inspection applications was predicted as qualified by the model, but it was actually unqualified.

**Table 4 foods-12-02118-t004:** Evaluation of imbalanced data sampling ratio.

Imbalanced Data Processing Methods and Sampling Ratio #	Precision (PPV)	Recall	F1
SMOTE 7:3	6.03%	64.91%	11.03%
SMOTE 6:4	5.68%	66.15%	10.46%
SMOTE 5:5	5.48%	75.16%	10.22%
SMOTE 4:6	4.94%	77.33%	9.28%
SMOTE 3:7	4.80%	81.68%	9.06%
Equal magnification 7:3	4.62%	87.89%	8.77%

Note #: The sampling ratio was 7:3 for qualified and unqualified products.

**Table 5 foods-12-02118-t005:** Index evaluation for 42 prediction models.

Data Set	Combination Number ^1^	Algorithm	ACR	Recall	PPV	F1	AUC	TN	FP	TP	FN	Sampling Rate	Rejection Rate
Validation set	A1	Bagging-C5.0	92.3%	2.2%	4.1%	2.8%	60.6%	2451	70	3	135	2.75	4.11
	A2	Bagging-CART	86.6%	25.4%	12.2%	16.5%	69.5%	2269	252	35	103	10.79	12.20
	A3	Bagging-EN	27.9%	90.6%	6.2%	11.5%	68.0%	617	1904	125	13	76.31	6.16
	A4	Bagging-GBM	84.7%	37.7%	13.9%	20.4%	72.3%	2200	321	52	86	14.03	13.94
	A5	Bagging-LR	83.0%	31.2%	10.8%	16.0%	68.0%	2164	357	43	95	15.04	10.75
	A6	Bagging-NB	69.7%	60.1%	9.9%	17.1%	73.2%	1769	752	83	55	31.40	9.94
	A7	Bagging-RF	93.4%	0.7%	2.6%	1.1%	71.0%	2483	38	1	137	1.47	2.56
	A8	EL	85.5%	28.3%	12.0%	16.8%	72.5%	2235	286	39	99	12.22	12.00
	B1	Bagging-C5.0	89.7%	22.5%	15.6%	18.4%	72.7%	2353	168	31	107	7.48	15.58
	B2	Bagging-CART	88.4%	19.6%	12.0%	14.9%	68.7%	2323	198	27	111	8.46	12.00
	B3	Bagging-EN	7.7%	97.8%	5.2%	9.9%	69.7%	71	2450	135	3	97.22	5.22
	B4	Bagging-GBM	90.1%	26.8%	18.6%	22.0%	73.1%	2359	162	37	101	7.48	18.59
	B5	Bagging-LR	87.9%	28.3%	14.9%	19.5%	71.2%	2299	222	39	99	9.82	14.94
	B6	Bagging-NB	79.6%	50.0%	12.7%	20.3%	73.3%	2048	473	69	69	20.38	12.73
	B7	Bagging-RF	90.6%	24.6%	18.9%	21.4%	75.2%	2375	146	34	104	6.77	18.89
	B8	EL	88.2%	31.9%	16.6%	21.8%	74.0%	2300	221	44	94	9.97	16.60
	C1	Bagging-C5.0	93.4%	11.6%	23.2%	15.5%	67.2%	2468	53	16	122	2.59	23.19
	C2	Bagging-CART	86.6%	39.9%	16.8%	23.6%	69.7%	2248	273	55	83	12.34	16.77
	C3	Bagging-EN	81.3%	11.6%	4.1%	6.0%	50.1%	2145	376	16	122	14.74	4.08
	C4	Bagging-GBM	88.9%	33.3%	18.5%	23.8%	73.0%	2318	203	46	92	9.36	18.47
	C5	Bagging-LR	86.1%	33.3%	14.2%	20.0%	69.0%	2244	277	46	92	12.15	14.24
	C6	Bagging-NB	72.5%	58.7%	10.7%	18.1%	73.7%	1847	674	81	57	28.39	10.73
	C7	Bagging-RF	94.6%	7.2%	38.5%	12.2%	75.4%	2505	16	10	128	0.98	38.46
	C8	EL	92.0%	23.2%	22.9%	23.0%	73.6%	2413	108	32	106	5.27	22.86
	D1	Bagging-C5.0	92.2%	21.7%	23.1%	22.4%	73.2%	2421	100	30	108	4.89	23.08
	D2	Bagging-CART	87.3%	28.3%	14.0%	18.8%	72.9%	2282	239	39	99	10.46	14.03
	D3	Bagging-EN	54.2%	50.7%	5.7%	10.3%	52.6%	1370	1151	70	68	45.92	5.73
	D4	Bagging-GBM	91.1%	20.3%	18.2%	19.2%	74.2%	2395	126	28	110	5.79	18.18
	D5	Bagging-LR	90.1%	22.5%	16.7%	19.1%	70.1%	2366	155	31	107	7.00	16.67
	D6	Bagging-NB	77.4%	52.2%	11.9%	19.3%	73.7%	1986	535	72	66	22.83	11.86
	D7	Bagging-RF	91.0%	29.0%	22.0%	25.0%	76.4%	2379	142	40	98	6.84	21.98
	D8	EL	90.2%	28.3%	19.4%	23.0%	75.1%	2359	162	39	99	7.56	19.40
	E1	Bagging-C5.0	92.3%	9.4%	14.1%	11.3%	66.3%	2442	79	13	125	3.46	14.13
	E2	Bagging-CART	84.8%	33.3%	12.9%	18.6%	68.4%	2210	311	46	92	13.43	12.89
	E3	Bagging-EN	88.2%	3.6%	2.7%	3.1%	58.1%	2341	180	5	133	6.96	2.70
	E4	Bagging-GBM	88.1%	27.5%	14.9%	19.3%	71.5%	2304	217	38	100	9.59	14.90
	E5	Bagging-LR	85.9%	32.6%	13.8%	19.4%	69.1%	2239	282	45	93	12.30	13.76
	E6	Bagging-NB	73.2%	58.0%	10.9%	18.3%	73.7%	1867	654	80	58	27.60	10.90
	E7	Bagging-RF	94.5%	2.9%	26.7%	5.2%	73.1%	2510	11	4	134	0.56	26.67
	E8	EL	91.1%	16.7%	15.9%	16.3%	70.9%	2399	122	23	115	5.45	15.86
	F1	Bagging-C5.0	92.4%	7.2%	11.9%	9.0%	71.1%	2447	74	10	128	3.16	11.90
	F2	Bagging-CART	89.9%	9.4%	8.3%	8.8%	67.2%	2377	144	13	125	5.90	8.28
	F3	Bagging-EN	62.3%	19.6%	2.9%	5.1%	55.6%	1629	892	27	111	34.56	2.94
	F4	Bagging-GBM	91.1%	16.7%	16.0%	16.3%	72.9%	2400	121	23	115	5.42	15.97
	F5	Bagging-LR	90.4%	18.8%	15.3%	16.9%	68.4%	2377	144	26	112	6.39	15.29
	F6	Bagging-NB	79.7%	47.8%	12.4%	19.6%	73.6%	2053	468	66	72	20.08	12.36
	F7	Bagging-RF	90.9%	17.4%	15.9%	16.6%	74.3%	2394	127	24	114	5.68	15.89
	F8	EL	91.7%	10.1%	12.5%	11.2%	72.1%	2423	98	14	124	4.21	12.50

Note ^1^: The data combination representation method was the time interval of data/whether blacklisted vendors are included/unbalanced data processing method. There were a total of 6 combinations, namely, A: 2016–2017/Yes/Proportional Amplification, B: 2016–2017/Yes/SMOTE, C: 2011–2017/Yes/Proportional Amplification, D: 2011–2017/Yes/SMOTE, E: 2011–2017/No/Proportional Amplification, and F: 2011–2017/No/SMOTE.

**Table 6 foods-12-02118-t006:** Index evaluation details of the optimal risk prediction model.

Data Set	Combination Number ^1^	Algorithm	ACR	Recall	PPV	F1	AUC	TN	FP	TP	FN	Sampling Rate	Rejection Rate
Test set	C1	Bagging-C5.0	94.7%	8.0%	15.7%	10.6%	68.0%	3921	70	13	150	2.00	15.66
	C2	Bagging-CART	89.4%	38.7%	15.6%	22.2%	71.6%	3649	342	63	100	9.75	15.56
	C3	Bagging-EN	88.9%	9.2%	4.6%	6.1%	53.2%	3678	313	15	148	7.90	4.57
	C4	Bagging-GBM	87.8%	39.9%	13.8%	20.5%	72.0%	3584	407	65	98	11.36	13.77
	C5	Bagging-LR	49.7%	64.4%	4.9%	9.1%	51.6%	1960	2031	105	58	51.42	4.92
	C6	Bagging-NB	74.3%	52.8%	8.0%	13.9%	66.8%	2999	992	86	77	25.95	7.98
	C7	Bagging-RF	95.8%	1.8%	18.8%	3.4%	72.5%	3978	13	3	160	0.39	18.75
	C8	EL	90.9%	31.9%	16.4%	21.6%	69.9%	3725	266	52	111	7.66	16.35
	D1	Bagging-C5.0	91.2%	12.3%	8.2%	9.8%	69.3%	3767	224	20	143	5.87%	8.20%
	D2	Bagging-CART	89.5%	14.7%	7.5%	9.9%	67.6%	3693	298	24	139	7.75%	7.45%
	D3	Bagging-EN	56.7%	52.1%	4.7%	8.6%	57.1%	2272	1719	85	78	43.43%	4.71%
	D4	Bagging-GBM	93.1%	13.5%	13.0%	13.3%	71.0%	3844	147	22	141	4.07%	13.02%
	D5	Bagging-LR	92.4%	16.6%	13.0%	14.6%	65.3%	3811	180	27	136	4.98%	13.04%
	D6	Bagging-NB	81.3%	39.9%	8.7%	14.3%	66.8%	3313	678	65	98	17.89%	8.75%
	D7	Bagging-RF	86.2%	33.1%	10.4%	15.8%	68.5%	3525	466	54	109	12.52%	10.38%
	D8	EL	91.4%	19.6%	12.3%	15.1%	69.0%	3763	228	32	131	6.26%	12.31%

Note ^1^: The representation of data combination was based on the time interval of data in year/whether blacklisted vendors are included/unbalanced data processing method. C: 2011–2017/Yes/proportionally, D: 2011–2017/Yes/SMOTE.

**Table 7 foods-12-02118-t007:** Evaluation of the prediction effectiveness of the optimal risk prediction model.

Data Year	Overall Sampling Inspection			EL V.2 Sampling Inspection				
	Number of Inspection Application Batches	Sampling Rate	Rejection Rate	Prediction Batch Number ^1^	Suggested Number of Inspection Batches	Sampling Rate	Number of Hit Batches	Hit Rate
2019	29,573	10.68%	2.09%	4154	318	7.66%	52	16.35%

Note ^1^: The predicted number of batches referred to the number of batches extracted from the 2019 border inspection application with sampling records and inspection results.

**Table 8 foods-12-02118-t008:** Differences between EL V.2 and EL V.1 modeling methods.

Model Differences	EL V.1	EL V.2	Description
Screening of characteristic risk factors	Single-factor analysis and stepwise regression were used to screen characteristic factors using simple statistical methods.	Elastic Net New data were added monthly to participate in modeling, and then key factors were selected for actual participation.	Prevent factor collinearity. Make the remaining factors more independent and important.
Add algorithms	5 algorithms	7 algorithms	When the prediction effect of multiple models is reduced, the AUC > 50% can still be retained for integration to improve the robustness of the model.
Adjust model parameters	F_β_ regulated the sampling inspection rate. Five models had consistent values.	F_β_ regulated the sampling inspection rate Seven models were independently adjusted.	The sampling rate was regulated, and the elasticity was set at 2–8%.

**Table 9 foods-12-02118-t009:** Analysis table of elastic F_β_ threshold regulation for each classification model.

Beta	PPV (or Sampling Inspection Rejection Rate)	Recall (or Hit Rate of Unqualified Products)	Number of Border Inspection Application Batches	Number of Sampling Inspection Batches	Sampling Rate	F_β_ Threshold				
						Bagging-NB	Bagging-C5.0	Bagging-CART	Bagging-LR	Bagging-RF
1	20.00%	33.33%	249	5	2.01%	0.94	0.76	0.48	0.46	0.68
1.2	20.00%	33.33%	249	5	2.01%	0.94	0.76	0.48	0.46	0.68
1.4	15.38%	66.67%	249	13	5.22%	0.94	0.46	0.48	0.46	0.64
1.6	21.43%	100.00%	249	14	5.62%	0.94	0.46	0.48	0.46	0.6
1.8	21.43%	100.00%	249	14	5.62%	0.94	0.46	0.48	0.46	0.6
2	21.43%	100.00%	249	14	5.62%	0.31	0.46	0.48	0.46	0.6
2.2	21.43%	100.00%	249	14	5.62%	0.31	0.46	0.48	0.46	0.6
2.4	17.65%	100.00%	249	17	6.83%	0.31	0.46	0.43	0.46	0.6
2.6	16.67%	100.00%	249	18	7.23%	0.31	0.33	0.43	0.46	0.6
2.8	16.67%	100.00%	249	18	7.23%	0.31	0.33	0.43	0.46	0.6
3	8.82%	100.00%	249	34	13.65%	0.31	0.33	0.28	0.12	0.6
3.2	5.45%	100.00%	249	55	22.09%	0.31	0.18	0.28	0.12	0.25

**Table 10 foods-12-02118-t010:** Analysis table of all classification models using fixed F_β_ threshold regulation.

Recommended Threshold	PPV Unqualified Rate of Sampling Inspection	Recall Identification Rate of Unqualified Sampling	Sampling Rate
0.39	13.29%	100.00%	9.29%
0.40	13.29%	100.00%	9.29%
0.41	14.79%	100.00%	8.41%
0.42	15.16%	100.00%	7.96%
0.43	15.45%	100.00%	7.56%
0.44	15.45%	100.00%	7.56%
0.45	20.43%	100.00%	6.19%

**Table 11 foods-12-02118-t011:** AUC comparison between EL V.1 and EL V.2 models.

Model Revision	AUC of Algorithm						
	Bagging- EN	Bagging- LR	Bagging-GBM	Bagging- BN	Bagging- RF	Bagging-C5.0	Bagging- CART
EL V.1	-	69.03%	-	53.43%	57.40%	63.20%	63.17%
EL V.2	63.39%	63.13%	62.67%	62.13%	61.41%	57.72%	49.40%

Note: EL is an abbreviation for Ensemble Learning.

**Table 12 foods-12-02118-t012:** EL V.1 and EL V.2 model prediction index evaluation table.

Year of Analysis	Model Revision	Number of Algorithms	Recall	PPV	F1
2020	EL V.1	5	25.00%	2.47%	4.49%
	EL V.2	7	58.33%	4.38%	8.14%

Note: EL V.1 analysis interval: 8 April 2020–2 August 2020; EL V.2 analysis interval: 3 August 2020–30 November 2020.

**Table 13 foods-12-02118-t013:** Statistical table for border inspection application and sampling over the years.

General Sampling Inspection Items for Each Year	Number of Inspection Application Batches	Overall Sampling Rate	EL Sampling Rate	Overall Rejection Rate	EL Rejection Rate
2022	27,074	10.90% (2952/27,074)	6.48% (1754/27,074)	3.01% (89/2952)	4.39% (77/1754)
2021	23,670	9.14% (2163/23,670)	6.24% (1478/23,670)	4.16% (90/2163)	6.36% (84/1478)
2020	26,823	6.07% (1629/26,823)	2.78% (745/26,823)	3.74% (61/1629)	5.10% (38/745)
2019	29,573	10.68% (3157/29,573)	-	2.09% (66/3157)	-

Note: On 8 April 2020, the general border sampling inspection for S-type food was adjusted from random sampling inspection to EL V.1 predictive sampling inspection. On 3 August 2020, it was converted to EL V.2 for prediction sampling inspection.

**Table 14 foods-12-02118-t014:** Statistical performance evaluation of the ensemble learning prediction model before and after its launch.

General Sampling Inspection and Evaluation Items for Each Year	Annual Overall Sampling Inspection		EL Sampling Inspection	
	Annual Rejection Rate (Number of Unqualified Pieces/Total Number of Sampled Pieces)	*p* Value	EL Rejection Rate (Number of EL Unqualified Pieces/Number of EL Sampled Pieces)	*p* Value
2022	3.01% (89/2952)	0.022 *	4.39% (77/1754)	0.000 ***
2021	4.16% (90/2163)	0.000 ***	6.36% (84/1478)	0.000 ***
2020	3.74% (61/1629)	0.001 **	5.10% (38/745)	0.000 ***
2019	2.09% (66/3157)		-	-

Note: The chi-square test was used to evaluate whether there was a significant impact on the evaluation results in the years before and after the launch (2019). “*” means *p* < 0.05; “**” means *p* < 0.01; “***” means *p* < 0.001.

## Data Availability

Data are contained within the article.

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
