# Peer review of "EL V.2 Model for Predicting Food Safety Risks at Taiwan Border Using the Voting-Based Ensemble Method"

_foods, 2023, doi:10.3390/foods12112118_

Round 1

Reviewer 1 Report

This paper aimed a second-generation ensemble learning pre-14 diction model, EL V.2, which was developed based on seven algorithms to enhance the "detection rate of 15 unqualified cases" and improve the robustness of the model. However, I would like to draw the attention of the authors to the important points that need to be corrected in the article.  the following points must be incorporated.

1.      Abstract needs to be more technical.

2.      The contribution is unclear. Refer to the following paper, and check how to specifically write the contribution at the end of the Introduction section. “An Investigation of Credit Card Default Prediction in the Imbalanced Datasets”.

1.      Clearly mention the research rap using the latest research on voting-based ensemble methods.

2.      The flow of the paper is not good.

3.      Consider the following paper to add in the literature review section: A Hybrid Approach to Tea Crop Yield Prediction Using Simulation Models and Machine Learning; Ensemble Learning Models for Food Safety Risk Prediction; A fuzzy inference-based decision support system for disease diagnosis; A Voting-Based Ensemble Deep Learning Method Focused on Multi-Step Prediction of Food Safety Risk Levels: Applications in Hazard Analysis of Heavy Metals in Grain Processing Products

4.      Add one comparative analysis to validate that the proposed technique significantly improved the performance.

5.      In section 2, the background of the research domain, the work of existing research, and the difference between the authors’ work and existing research should be stated clearly in the literature review.

6.      English grammar needs to be checked.

7.      Authors should use some hypothesis testing to prove the developed model is significantly the same or different.

8.      Add the discussion section after the results to prove the significance of the study.

9.      Overall speaking, the innovation points and main contributions of this paper need to be carefully reconsidered, and the innovation points should be presented more clearly and more prominent in terms of word expression and methodology & experiment design.

Reviewer 2 Report

Provide detailed descriptions of the algorithms: The paper should provide detailed descriptions of the five algorithms used in EL V.1 and the seven algorithms used in EL V.2, including how they were chosen and how they were combined.

Specify hyperparameters: The paper should specify the hyperparameters used in the algorithms, such as learning rates and regularization parameters, to ensure that the models can be replicated by other researchers.

Provide more information on the dataset: The paper should provide more information on the dataset used to train and test the models, such as the size, distribution, and types of data, to help readers understand the limitations and generalizability of the models.

Clarify the confusion matrix: The paper should clarify how the confusion matrix was used to evaluate the prediction effects of EL V.1 and EL V.2, including how the true positive, true negative, false positive, and false negative rates were calculated.

Discuss the trade-off between accuracy and efficiency: The paper should discuss the trade-off between accuracy and efficiency in border food management, including how the models can balance the need for accurate risk assessment with the need for timely inspection.

Provide a more detailed discussion of the results: The paper should provide a more detailed discussion of the results, including potential reasons for the higher unqualified rates in 2020, 2021, and 2022 compared to 2019, and potential implications of the superior predictive performance of EL V.2 compared to EL V.1.

Please avoid using references that were published before 2018. Cite cutting-edge papers that are extremely relevant to your topic. Additionally, the paper lacks enough citations. Another crucial step is to discuss the paper's subject with other cutting-edge works or similar works in the introduction part to expand the paper's implications beyond the subject. These significant works may be referenced and used by authors to discuss their paper's subject and today's challenges.

Adak, A.; Pradhan, B.; Shukla, N. Sentiment Analysis of Customer Reviews of Food Delivery Services Using Deep Learning and Explainable Artificial Intelligence: Systematic Review. Foods 2022, 11, 1500. https://doi.org/10.3390/foods11101500

A. Heidari, N. J. Navimipour, M. Unal, and G. Zhang, "Machine Learning Applications in Internet-of-Drones: Systematic Review, Recent Deployments, and Open Issues," ACM Comput. Surv., vol. 55, no. 12, p. Article 247, 2023, doi: 10.1145/3571728.

A. Heidari, M. A. J. Jamali, N. J. Navimipour and S. Akbarpour, "A QoS-Aware Technique for Computation Offloading in IoT-Edge Platforms Using a Convolutional Neural Network and Markov Decision Process," in IT Professional, vol. 25, no. 1, pp. 24-39, Jan.-Feb. 2023, doi: 10.1109/MITP.2022.3217886.

Daniels, J.; Herrero, P.; Georgiou, P. A Deep Learning Framework for Automatic Meal Detection and Estimation in Artificial Pancreas Systems. Sensors 2022, 22, 466. https://doi.org/10.3390/s22020466

Minor editing of the English language required

Reviewer 3 Report

The topic is important both from a theoretical perspective and from a very concrete, practical point of view. 

Main recommendations:

1. I appreciate that the introduction provides information about the size of the problem, but I would recommend removing Figure 1 from that section. It is unusual to have pictures in the introductory part, and in addition I think the numbers provided on lines 32-33 are more than enough to provide the reader with a good understanding of the importance of the imported food. Or maybe include Figure 1 in the appendix?

2. I would welcome a bit more information about the type of risks the border faces in respect to the quality of the food. Can you give examples of high-risk products? 

3. Section 2.1. does not seem to belong to the Literature review section. The information here presents the context, and not the state of the art in relation to your research question. Maybe it needs to be reorganized and included somewhere else. 

4. In a similar vein, section 2.2. does not belong to the Literature review section, but to a separate section usually entitled Data and methods.

5. Section 2.3. seems to be misplaced too, as it provides more of a context, rather than a literature review. Lines 112- 117 provide the first information about a research gap, but it does not provide any references, and seems to be disconnected with the aim of the paper.

Lines 115 - 117 announce that the goal of the paper is to effectively checking the quality of the food, while in the introductory part the aim of the paper is to improve an existing algorithm. In which way, and why is this existing algorithm in need for improvements? What are its current failures? Why do you want to develop a better one, and what are the current failures that you aim to address?

I recommend a better and clearer focus on the research gap the authors aim to fill. The literature review should be build around the research question, which in this case is fuzzy and poorly stated. 

6. Section 2.4.: In which way is this section related to your paper? Do the algorithms discussed in this section relate with the one used in Taiwan? Literature review is not a list of topics related to the keywords, but building blocks that support the research question. 

7. Section 2.5. You write about the USA: is this information related with your paper? If yes, in which way? What is the use of sections 2.4. and 2.5. in the architecture of the paper? In which way the algorithm used in Taiwan relates with the ones developed in Europe and the USA? 

On lines 166 - 168 you write: "Therefore, this study referred to the data sources and practices of the European Union and the United States to collect risk factors and establish prediction model planning.". However, there is no reference in the entire section to any risk factors - but only to methods employed to detect low-quality food. 

Once again, I recommend to reconsider the research question, make it clearer, and develop the literature review around the research gap the authors aim to fill. 

I am not qualified to judge in detail the method, and the results. 

English language is fine, or at least this is how it seems to me. Being a non-native myself, I am not bothered by many of the details a native would perceive as inappropriate. 

Reviewer 4 Report

·         Define abbreviations in abstract.

·         The title needs to be more specific. What do you mean by improved model?  Specify the model

·         The statistical results in abstract, Line 20, is confusing. Why so many 0? Can this be removed??

·         Fig 1 is simple and maybe can be described in the text. The authors can reconsider again.

·         List of abbreviations/acronyms can be provided.

·         The objectives of works seem unclear. Please specify before the method

·         Line 208, why only S-type products? Can you detail this?

·         Specify the statistical software employed in your study? The version as well

·         The models employed needs citations. The ones original before the improvement was made.

·         Table 4, the combination numbers, what are those?

·         Do you have the standard error values?

·         For the research limitations, what would be your suggested solutions for the future to overcome those limitations?

·         Overall the paper is very comprehensive.

Please proofread for final round.

Round 2

Reviewer 1 Report

Accepted.

Reviewer 2 Report

No comment.

Reviewer 3 Report

I am fine with the revisions. 

English language seems to be fine, minor revisions are required.